# Integrating Metabolomics and Network Pharmacology to Explore the Mechanism of Xiao-Yao-San in the Treatment of Inflammatory Response in CUMS Mice

**DOI:** 10.3390/ph16111607

**Published:** 2023-11-14

**Authors:** Yi Zhang, Xiao-Jun Li, Xin-Rong Wang, Xiao Wang, Guo-Hui Li, Qian-Yin Xue, Ming-Jia Zhang, Hai-Qing Ao

**Affiliations:** 1Department of Psychology, School of Public Health and Management, Guangzhou University of Chinese Medicine, Guangzhou 511400, China; zybzy7567@163.com (Y.Z.); wangxinronggucm@163.com (X.-R.W.); liguohuixs@163.com (G.-H.L.); xqy777777@163.com (Q.-Y.X.); 2School of Chinese Pharmaceutical Science, Guangzhou University of Chinese Medicine, Guangzhou 511400, China; xiaojunli25@163.com; 3Department of Basic Theory of TCM, Guangzhou University of Chinese Medicine, Guangzhou 511400, China; 18487288184@163.com; 4Department of Basic Theory of TCM, Zhejiang Chinese Medical University, Hangzhou 310000, China

**Keywords:** Xiao-Yao-San, CUMS, depression, inflammation, network pharmacology, metabolomics

## Abstract

Depression can trigger an inflammatory response that affects the immune system, leading to the development of other diseases related to inflammation. Xiao-Yao-San (XYS) is a commonly used formula in clinical practice for treating depression. However, it remains unclear whether XYS has a modulating effect on the inflammatory response associated with depression. The objective of this study was to examine the role and mechanism of XYS in regulating the anti-inflammatory response in depression. A chronic unpredictable mild stress (CUMS) mouse model was established to evaluate the antidepressant inflammatory effects of XYS. Metabolomic assays and network pharmacology were utilized to analyze the pathways and targets associated with XYS in its antidepressant inflammatory effects. In addition, molecular docking, immunohistochemistry, Real-Time Quantitative Polymerase Chain Reaction (RT-qPCR), and Western Blot were performed to verify the expression of relevant core targets. The results showed that XYS significantly improved depressive behavior and attenuated the inflammatory response in CUMS mice. Metabolomic analysis revealed the reversible modulation of 21 differential metabolites by XYS in treating depression-related inflammation. Through the combination of liquid chromatography and network pharmacology, we identified seven active ingredients and seven key genes. Furthermore, integrating the predictions from network pharmacology and the findings from metabolomic analysis, Vascular Endothelial Growth Factor A (VEGFA) and Peroxisome Proliferator-Activated Receptor-γ (PPARG) were identified as the core targets. Molecular docking and related molecular experiments confirmed these results. The present study employed metabolomics and network pharmacology analyses to provide evidence that XYS has the ability to alleviate the inflammatory response in depression through the modulation of multiple metabolic pathways and targets.

## 1. Introduction

Depression, also known as depressive disorder, is a highly prevalent clinical psychiatric condition [1]. Multiple studies have confirmed that the development of depression can lead to immune dysfunction and trigger an inflammatory response in the body [2]. Several meta-analyses have found that pro-inflammatory cytokines, such as C-reactive protein (CRP), interleukin-6 (IL-6), tumor necrosis factor-alpha (TNF-α), and interleukin-6 (IL-1β), are elevated in depressed patients compared to normal populations [3,4,5]. The development of depression can also exacerbate other inflammation-related diseases, such as obesity, asthma, rheumatoid arthritis, and cardiovascular disease [6,7]. Although existing antidepressants show promise in reducing peripheral inflammation in patients with evidence of depression and in animal models of stress, there is currently a lack of approved drugs for antidepressant treatment that modulate the inflammatory response [8]. It is crucial to discover antidepressants that are both safe and effective in regulating the inflammatory response to impede the progression of the disease.

Traditional Chinese herbal medicine is becoming increasingly popular for its low side effects and high safety. Xiao-Yao-San (XYS), recorded in the “Tai Ping Hui Min He Ji Ju Fang” over a thousand years ago, has been used in Traditional Chinese Medicine (TCM) to treat depression [9]. XYS is composed of eight herbs, including *Angelica sinensis*, *Paeonia lactiflora*, *Bupleurum chinense*, *Poria cocos*, *Zingiber officinale*, and *Mentha haplocalyx*. According to TCM theory, XYS has the effects of soothing the liver and tonifying the spleen. Many studies have already proved that XYS has a good anti-depressive effect [10]. Furthermore, a study found that Xiaoyao Pills improved depressive-like behavior induced by Lipopolysaccharide (LPS), and also reduced the levels of peripheral serum inflammatory factors IL-6 and TNF-α [11]. However, there is limited research on the anti-inflammatory effects of XYS, as well as its potential to improve the inflammatory response in depressive states, and the mechanisms behind these effects are still unclear.

In recent years, with the innovation of research theory and technology, investigating the mechanisms of Traditional Chinese Medicine in treating diseases through techniques such as network pharmacology and metabolomics has become a research hotspot. Metabolomics involves analyzing the subtle changes in the biological pathways of metabolites in the body to understand the metabolic mechanisms of various physio-pathological processes and drug treatments [12]. Network pharmacology, meanwhile, uses drug-target-disease analysis to predict the molecular mechanisms and pathways of drug action in diseases [13]. When combined, these two techniques can provide a more in-depth understanding of the role of TCM in therapeutics, from active ingredients to disease targets, metabolites, and pathways. This approach bridges the gap between the lack of experimental validation in network pharmacology and the lack of upstream molecular mechanisms and drug binding targets in metabolomics. As a result, the study of TCM through these innovative techniques has become a hot research topic.

Chronic psychological stress plays a crucial role in the development of various psychiatric disorders, including depression. Animal models like chronic unpredictable mild stress (CUMS) are regularly employed to mimic depressive behavior [14]. In this research, CUMS model mice were utilized to investigate the mechanism by which XYS regulates inflammation caused by chronic psychological stress, employing a combination of metabolomics, network pharmacology, and experimental validation. Figure 1 represents the experimental flowchart of this study. This study offers insights into the clinical application and mechanistic study of XYS.

## 2. Results

### 2.1. Quality Control of XYS by Liquid Chromatography Tandem Mass Spectrometry (LC-MS/MS)

In order to identify the major chemical components of XYS, LC-MS/MS analysis was conducted on the sample. The total ion chromatograms (TIX) of XYS in positive (Figure 2a) and negative (Figure 2b) modes displayed the chemical composition of all compounds. In total, nine chemical components were identified. After reviewing the Traditional Chinese Medicine Systems Pharmacology Database and Analysis Platform (TCMSP) and literature, Gingerol was excluded due to its low oral bioavailability. Ultimately, eight main active ingredients were included: Paeoniflorin, Liquiritin, Ferulic acid, Atractylenolide I, 2-Atractylenolide, Atractylenolide III, Saikosaponin A, and Saikosaponin D (Table 1).

### 2.2. Effect of Xiao-Yao-San on Depression-like Behaviors in Chronic Unpredictable Mild Stress Mice

The Open Field Test (OFT) (Figure 3b–d), Sucrose Preference Test (SPT) (Figure 3e), Forced Swimming Test (FST) (Figure 3f), and Tail Suspension Test (TST) (Figure 3g) were utilized to evaluate behavioral variations resulting from CUMS treatment. The CUMS group exhibited a reduced total walking distance, total walking time, and central activity distance during the OFT in comparison to the control group (*p* < 0.01). However, treatment with XYS-L, XYS-M, and XYS-H effectively reversed these alterations induced by CUMS (*p* < 0.01, *p* < 0.01, *p* < 0.01). In comparison to the control group, the CUMS-treated mice showed a significant decrease in their sugar-water preference rate (*p* < 0.01).

However, the XYS-M and XYS-H groups displayed an increase in sugar-water consumption in contrast to the CUMS group (*p* < 0.05, *p* < 0.01). No significant difference in sugar-water consumption was observed in the XYS-L group. Furthermore, the CUMS group exhibited longer immobility times in both the Forced Swimming Test (FST) (*p* < 0.01) and Tail Suspension Test (TST) (*p* < 0.01) compared to the control group. Conversely, administration of XYS-L, XYS-M, and XYS-H significantly reduced the immobility times in the FST (*p* < 0.01, *p* < 0.01, *p* < 0.01) and TST (*p* < 0.01, *p* < 0.01, *p* < 0.01). The above observations indicate the establishment of a depression model using CUMS, and XYS exhibited potential for reversing depression-like behavior.

### 2.3. Effect of Prolotherapy on the Inflammatory Response in the Spleen of CUMS Mice

To investigate the potential immunomodulatory effects of XYS in a chronic unpredictable mild stress (CUMS) model, the serums were subjected to Enzyme Linked Immunosorbent Assay (ELISA) analysis to evaluate the expression levels of IL-6, TNF-α, and IL-1β (Figure 4a–c). In comparison to the control group, the CUMS model mice showed significantly increased levels of IL-6, TNF-α, and IL-1β (*p* < 0.01, *p* < 0.01, *p* < 0.01), along with a higher splenic index (Figure 4d) (*p* < 0.01), which indicates the presence of an inflammatory response in the CUMS model. However, treatment with XYS-M and XYS-H resulted in significantly lower expression levels of IL-6 (*p* < 0.01, *p* < 0.01), TNF-α (*p* < 0.05, *p* < 0.01), and IL-1β (*p* < 0.01, *p* < 0.01), and a reduction in the splenic index (*p* < 0.05, *p* < 0.05) compared to the CUMS group. These results strongly suggest that XYS may effectively alleviate the inflammatory response that is induced by chronic psychological stress.

### 2.4. XYS Improved the Metabolic Profile of the Inflammatory Response in CUMS Mice

We selected the control and CUMS groups, as well as the drug concentration group with the most pronounced modeling effect, the XYS-H group, for metabolomic analysis. We utilized an Ultra-High Performance Liquid Chromatography Tandem Mass Spectrometry (UHPLC-MS/MS) system to conduct metabolomic analysis of spleen tissue. To evaluate the quality of the metabolomic data analysis, we employed unsupervised principal component analysis (PCA). The results indicated that the quality control (QC) samples were closely clustered together at the center of the three sample groups, demonstrating the precision of our metabolic data. Furthermore, the distinct clustering patterns observed among the control, CUMS, and XYS-H groups indicate that this metabolomic dataset exhibits significant differences between these groups (Figure 5a,b).

Furthermore, orthogonal partial least squares discriminant analysis (OPLS-DA) was employed to further investigate the differences between the groups. The R2X (cum) and Q2 (cum) values were utilized to assess the fitting of the CUMS to the data. The parameters for the Control group versus the CUMS group in the ESI+ (Figure 5c) and ESI− (Appendix A) modes were R2X (0.851, 0.782) and Q2 (0.989, 0.923), respectively. Additionally, for the CUMS versus XYS-H groups, the R2X (0.913, 0.862) and Q2 (0.988, 0.986) values were obtained in the ESI+ (Figure 5f) and ESI− (Appendix A) modes. These findings indicate the effective establishment of the model. To verify the accuracy and validity of the data analysis, a permutation test was performed 200 times. The cross-validation plots demonstrated that the multivariate data analysis was not over-fitted, and the model exhibited excellent predictability (Figure 5d,g and Appendix A).

Based on OPLS-DA analysis, potential differential metabolites were screened based on a Variable Importance in Projection (VIP) > 1.0 and *p* < 0.05 using referenced criteria. Using the Human Metabolome Database (HMDB) and Metlin, 21 potential differential metabolites were initially identified (Figure 6a) (Table 2). Out of these, seven metabolites were found to be down-regulated and 14 metabolites were up-regulated in the CUMS group compared to the control group, as presented in the table. To further identify the pathways of action of these differential metabolites, metabolite pathway analysis was conducted using MetaboAnalyst 5.0. This analysis screened out metabolic pathways with *p* < 0.05 and Impact > 0, resulting in the identification of four relevant metabolic pathways (Figure 6b) (Table 3): Pyrimidine metabolism, Glutathione metabolism, Phenylalanine, tyrosine and tryptophan biosynthesis, and Glycine, serine, and threonine metabolism.

### 2.5. Network Pharmacology Analysis

By integrating the compounds in TCMSP with identified compounds, a total of 126 compounds and 320 relevant targets were obtained. The Genecard, Therapeutic target database (TTD), Drugbank, and Online Mendelian Inheritance in Man (OMIM) databases were searched for disease targets associated with “depression” and “inflammation”, resulting in the identification of 1111 and 1477 disease targets, respectively. Jvenn was then used to obtain intersections of depression, inflammation, and XYS target genes, resulting in 66 XYS genes that may act on the inflammatory response of CUMS (Figure 7a) (Appendix A).

A PPI network was constructed using the STRING database with intersection targets (Figure 7b), consisting of 66 nodes and 818 edges, with an average node degree of 24.79. Cytoscape (version 3.8.2) was employed for visualizing the network. To identify significant targets, topological analysis was conducted using the Cytoscape Network Centrality Analysis (CytoNCA) plug-in. This yielded the following Hub targets that met the criteria of “twice the median of Degree ≥ 44, average Betweenness ≥ 43.88, average Closeness ≥ 0.61”: RAC-alpha serine/threonine-protein kinase (AKT1), Vascular endothelial growth factor A (VEGFA), Interleukin-1 beta (IL-1β), Cellular tumor antigen p53 (TP53), Caspase-3 (CASP3), Peroxisome proliferator-activated receptor gamma (PPARG), and Prostaglandin G/H synthase 2 (PTGS2) (Table 4) (Figure 7c).

To further investigate the mechanism of action of XYS in reversing the inflammatory response in CUMS, we utilized Metascape for the Kyoto Encyclopedia of Genes and Genomes (KEGG) pathway and Gene Ontology (GO) enrichment analysis on 66 intersecting targets. The top 20 pathways were visualized (Figure 7d) and the results revealed significant effects in the HIF-1 signaling pathway, IL-17 signaling pathway, PI3K-Akt signaling pathway, and JAK-STAT signaling pathway, all of which play roles in stress and inflammatory responses. The GO analysis (Figure 7e) highlights the importance of biological processes (BP) such as the response to lipopolysaccharide, molecules of bacterial origin, xenobiotic stimulus, bacterium, oxygen levels, inorganic substance, extracellular stimulus, and cellular response to lipid. It also emphasizes molecular functions (MF) such as signal receptor regulator activity, protein homodimerization activity, signaling receptor activator activity, cytokine receptor binding, and growth factor receptor binding. These findings suggest that XYS may act on the inflammatory response of CUMS through multiple pathways and targets. Through an extensive search of the Human Metabolome Database (HMDB), 210 protein targets linked to potential metabolites were obtained (Appendix A).

### 2.6. Molecular Docking

We integrated network pharmacology predictions with metabolomic analysis to identify two key targets, namely VEGFA and PPARG. Using Cytoscape’s plugin CytoNCA, we conducted topological analysis of the disease-ingredients-target network map (Figure 7f), which resulted in identifying seven key components (Table 5). Furthermore, molecular docking of the screened core targets and the hub components was performed. The lower the energy at which the ligand binds to the receptor in a stable conformation, the higher the likelihood of an effect occurring. The docking results (shown in Figure 8a) revealed 14 sets of receptor-ligand docking results. The binding scores for the potential targets range from −9.6 kcal·mol^−1^ to −6.2 kcal·mol^−1^ with an average binding energy of −7.98 kcal·mol^−1^, indicating strong binding activity between the core components of XYS and the predicted key targets.

### 2.7. The Role of Key Targets of XYS in Treating Inflammatory Responses to Chronic Psychological Stress

To further validate the accuracy of the predicted key targets, we conducted immunohistochemical (Figure 9a,c,d), RT-qPCR (Figure 9e,f), and Western Blot (Figure 9b,g,h) assays on VEGFA and PPARG in mouse spleen tissues. The results revealed a significant elevation in the expression of VEGFA in the CUMS model group compared to the control group, as confirmed by both immunohistochemical, RT-qPCR, and Western Blot analyses. In contrast, the treatment with XYS effectively reversed the abnormal upregulation of VEGFA in the CUMS group. Conversely, the expression of PPARG was markedly suppressed in the CUMS group compared to the control group, whereas it was significantly elevated in the XYS-H group compared to the CUMS group.

## 3. Discussion

Repeated, continuous, or excessive exposure to stress can result in an inflammatory state [7]. Studies have shown that persistent or recurrent stressful events can activate and increase inflammatory cells in the peripheral blood or tissues of the body [15], leading to elevated levels of inflammatory markers such as interleukin-6, interleukin-1β, C-reactive protein, and tumor necrosis factor, among others, even in the absence of a causative agent [3,5]. This, in turn, may increase the probability of developing infectious diseases or autoimmune disorders.

The findings of this study suggest that CUMS modeling can induce organismal inflammation, which is consistent with previous research. However, XYS was found to inhibit the expression of inflammatory cytokines in CUMS mice. To further understand the underlying mechanism, we conducted a metabolomic analysis of the spleen, which is the largest lymphoid organ in the body. This analysis identified 21 differential metabolites. The main enriched pathways included pyrimidine metabolism, glutathione metabolism, phenylalanine, tyrosine and tryptophan biosynthesis, and glycine, serine, and threonine metabolism. Research has found an enhanced state of oxidative stress in both depressive symptoms and inflammatory responses [16]. Glutathione, the most abundant sulfur-hydrogen compound in animals, plays a crucial role in combating the oxidative stress response within the organism [17]. In this experiment, it was observed that the CUMS group exhibited an increase in oxidized glutathione levels, indicating an enhanced oxidative stress response in the organism. Phenylalanine, tyrosine, and tryptophan play crucial roles in the development of depression [18]. According to the “monoamine deficiency hypothesis”, depression is linked to a reduction in the brain neurotransmitters serotonin and norepinephrine [19]. Tryptophan, which acts as a precursor to serotonin, can be converted to serotonin through oral intake and has sedative and sleep-inducing effects [20]. Phenylalanine can be converted to tyrosine, and tyrosine hydroxylation produces norepinephrine [21]. Depletion of the coenzyme tetrahydrobiopterin (BH4) is necessary for this conversion process [22]. In patients with mood disorders, chronic inflammation occurs in the body, and cytokines induce the production of reactive oxygen species, leading to the depletion of BH4 [22,23,24]. This depletion in turn reduces the conversion of norepinephrine. Glycine is an inhibitory neurotransmitter produced by mammals that serves various functions in neural and peripheral tissues, including its roles as an antioxidant, anti-inflammatory agent, and an immunomodulator [25,26].

Combining metabolomics and network pharmacology, VEGFA and PPARG were the key targets of the antidepressant inflammatory response of XYS. Molecular docking confirmed the strong binding activity between the active ingredients of XYS and the key targets. VEGFA is a major pro-angiogenic factor expressed in human neurons, astrocytes, and vascular endothelial cells, which can exert powerful neurotrophic effects [27]. Some studies have revealed that there is an elevated expression level of VEGFA in the plasma or serum of patients with Major Depressive Disorder (MDD) [28]. This suggests that VEGFA has the potential to serve as a biomarker for MDD [29]. However, the secretion of VEGFA by astrocytes can negatively impact the expression of transmembrane tight junction proteins claudin-5 (CLN-5) and occludin (OCLN) within the central nervous system (CNS) in vivo. This disruption can induce the breakdown of the blood–brain barrier (BBB) catabolism, which in turn facilitates the infiltration of inflammatory factors and immune cells, ultimately resulting in inflammation [30]. In the present study, it was observed that the administration of XYS resulted in a reduction in VEGFA expression levels. This finding indicates that XYS has the potential to reduce vascular permeability by inhibiting the overexpression of VEGFA. As a result, peripheral inflammatory factors and cell aggregation could be reduced.

Peroxisome proliferator-activated receptor-γ (PPARG), originally identified in sawfish, belongs to the nuclear receptor superfamily of ligand-activated transcription factors [31]. It plays a crucial role in various aspects, including insulin sensitivity, lipid and glucose metabolic homeostasis, as well as inflammation regulation [32]. Festuccia et al. discovered that PPARG agonists have the ability to hinder neurotoxicity triggered by excessive norepinephrine and glucocorticoid production, while promoting nerve growth during neuronal stress [33]. Research has indicated that the PPARG agonist pioglitazone may have the potential to decrease the severity of depression [34]. It has been demonstrated to inhibit the expression of TNF-α, IL-6, and IL-1β in monocytes. PPARG exerts negative regulation on the transcription of inflammation-responsive genes by antagonizing the activator protein-1 and NF-kB signaling pathways [35]. Therefore, the inflammatory response triggered by stress can potentially result in a compensatory reduction in PPARG levels within the body. In this particular experiment, mice in the CUMS group exhibited increased expression of IL-6, TNF-α, and IL-1β, along with decreased expression of PPARG, aligning with the outcomes of previous studies. These findings suggest that XYS has the potential to ameliorate stress and inflammatory response through the enhancement of PPARG.

## 4. Materials and Methods

### 4.1. Chemicals and Reagents

XYS is composed of *Angelica Sinensis Radix granules* (Lot No. 00449291), *Paeoniae Radix Alba granules* (Lot No. 8121031), *Bupleuri Radix granules* (Lot No. 1091641), *Poria granules* (Lot No. 0100221), *Zingiberis Rhizoma Recens granules* (Lot No. A1072551), *Menthae Haplocalycis herba granules* (8110811), *Atractylodis Macrocephalae Rhizoma granules* (Lot No. 9039201), and *Glycyrrhizae Radix Et Rhizoma Praeparata Cum Melle* (Lot No. 1019421), all of which were purchased from Efonge Pharmaceuticals (Guangzhou, China). Formic acid, methanol, and acetonitrile (UPLC-MS grade) were purchased from Comio Chemical Reagents Ltd. (Tianjin, China). The BCA protein quantification kit was purchased from Biyuntian Biotechnology (Shanghai, China). The mouse TNF-α, IL-6, and IL-10 ELISA kits were obtained from Jiangsu Meimian Industrial CO., Ltd. (Yancheng, China).

### 4.2. Animal Experiment

Male c57BL/6 mice (18–22 g) (Animal license No. SCXK2019-0047) were obtained from the Experimental Animal Center of Guangzhou University of Chinese Medicine (Guangzhou, China). The mice were housed in a controlled environment with the temperature at 22 ± 2 °C, humidity at 60 ± 5%, and a 12 h/12 h day/night cycle. All animal experiments were conducted in accordance with the Ethics Committee of Guangzhou University of Traditional Chinese Medicine (Guangzhou, China) and were registered under the number 20210604001. After a week of acclimatization, 50 mice were randomly allocated into five groups: control, chronic unpredictable mild stress (CUMS), CUMS+ low-dose XYS (XYS-L,1.4 g/kg/d), CUMS+ medium-dose XYS (XYS-M, 2.8 g/kg/d), and CUMS+ high-dose XYS (XYS-H, 5.6 g/kg/d). The dose of XYS in the XYS-L group mice was calculated by converting it based on the standard human clinical dosage and the administration dosage conversion coefficient from human to mouse (the administration dosage in the XYS-L group = 9.1 × human drug dosage/60 kg) (1.4 g/kg/d). Subsequently, the medium dose (2.8 g/kg/d) and high dose (5.6 g/kg/d) were considered as two-fold and four-fold of the initial dose, respectively). The control and model groups were administered an equivalent volume of saline water by intragastric gavage. Additionally, the mice received drug treatment continuously during the five-week CUMS procedure.

#### 4.2.1. *CUMS Modeling Methods*

CUMS modeling methods included: (1) 45° inclined cage for 24 h; (2) restraint for 6 h; (3) food deprivation for 24 h; (4) water deprivation for 24 h; (5) solitary confinement for 12 h; (6) 4 °C cold water swimming for 5 min; (7) 300 mL water/cage wet cage housing for 20 h; (8) light/dark recycle reversal for 24 h. Two types of stimulation were selected each day, and stimulation was not repeated for two consecutive days for five weeks [15]. At the end of the modeling process, behavioral experiments such as Open Field, Forced Swimming, and Sucrose Preference were conducted. Mouse moulding time flow is shown in Figure 3a.

The mice were subjected to a 12-h fast before sampling and their weight was recorded. For anesthesia, sodium pentobarbital was administered through injection. The blood samples were obtained by removing eyeballs, and the serums were collected by centrifugation at 2000× *g* for 20 min at 4 °C. The spleens of each group of mice were collected and weighed. Afterwards, they were either fixed in 4% neutral formaldehyde or frozen for storage. The spleen index was calculated with the following formula:Spleen index = spleen weight (mg)/body weight (g)

#### 4.2.2. Behavioral Testing

##### Sucrose Preference Test (SPT)

On day 35 of modeling, sugar-water preference experiments were conducted as follows [36]: mice were individually housed and provided with two 1% sucrose solution bottles for 24 h to facilitate their acclimation to the sugar solution. Afterward, their water bottles were replaced with a bottle of 1% sucrose solution and another of purified water for 24 h. Following a 24-h period of food and water deprivation, mice were subjected to sugar-water preference tests, during which they were given unrestricted access to the two bottles. After 12 h, the weight of each bottle was recorded, and the sucrose-water preference rate for each mouse was calculated using the formula: sucrose preference = (sucrose consumption/(water consumption + sucrose consumption)) × 100%.

##### Open Field Test (OFT)

The OFT was utilized to evaluate movement and exploratory behaviors. As per the previous protocol [37], open-field experiments were conducted at the conclusion of the modeling process. Each mouse was placed individually in a 50 × 50 × 40 white background box and allowed to move freely for a duration of seven minutes. A camera was utilized to record the mouse’s behavior during the test. Behavioral instruments were employed to measure the total walking distance, total walking time, and central distance traveled by the mice during the test. Additionally, the OFT box was disinfected with 75% alcohol prior to the next mouse test (Shanghai Jiliang Software Technology, Shanghai, China).

##### Forced Swimming Test (FST)

The Forced Swimming Test (FST) was conducted to assess helplessness, following established procedures [36]. Each mouse was placed in a clear cylindrical container (10 cm diameter, 25 cm height) filled with 15 cm of water (24 ± 1 °C) and tested for 7 min. The mouse’s swimming behavior was recorded using a camera and analyzed during the last 5 min of immobility using Shanghai Jiliang Software Technology (China).

##### Tail Suspension Test (TST)

TST was frequently employed to evaluate depressive behavior, similar to prior research [37]. During the experiment, the rear one-third of the mouse’s tail was taped and suspended 40 cm above the ground for 7 min. A horizontal camera was utilized to capture the procedure, and the duration of immobilization over five minutes was analyzed using Shanghai Jiliang Software Technology, China.

### 4.3. Preparation and Compound Identification of Xiao-Yao-San (XYS)

#### 4.3.1. Preparation of Xiao-Yao-San

To prepare the XYS extract, granules of Angelica Sinensis Radix, Paeoniae Radix Alba, Bupleuri Radix, Poria, Zingiberis Rhizoma Recens, Menthae Haplocalycis herba, Atractylodis Macrocephalae Rhizoma, Glycyrrhizae Radix Et Rhizoma Praeparata Cum Melle were mixed in the ratio of 2:2:2:2:2:1:1:1 [38]. The drug was added to an appropriate amount of pure water according to the number of the administered mice, fully dissolved, heated and boiled for 25 min, filtered through a 25 µm cartridge, and analyzed by liquid chromatography tandem mass spectrometry (LC-MS/MS).

#### 4.3.2. Compound Composition of Xiao-Yao-San

LC-MS/MS analysis of the drug was performed using SCIEX Triple T of 5600. The analysis was performed using an Agilent Poroshell column (100 × 3.0 mm, 2.7 µm). Mobile phase A comprised 0.1% formic acid in the water, and mobile phase B was composed of acetonitrile. Separation and identification were performed using a gradient elution procedure as follows: 0–12 min, 5–20% B; 12–18 min; 40–70% B; 20–22 min, 70–5% B. The injection volume was 2 µL. The column temperature was kept at 35 °C and the flow rate was 0.2 mL/min. Chromatography and mass spectrometry detection were carried out in positive ion mode. The ion source temperature was 550 °C; the ion source gas1 (gas1) pressure was 1:50 psi (1 psi = 6.895 kPa); the ion source gas2 (gas2) pressure was 55 psi; the curtain gas pressure was 35 psi; and the floating Ionspray voltage was 5.5 kV. The Analyst TF 1.8 Software acquisition system and SCIEX OS were used for data acquisition and data analysis, respectively.

### 4.4. Metabolomic

#### 4.4.1. Sample Preparation

The spleen tissue was ground into a fine powder using liquid nitrogen and transferred into EP tubes. A 500 µL solution of 80% methanol in water was added to each tube, and the mixture was vortexed thoroughly before being placed in an ice bath for 5 min. The tubes were then centrifuged at 15,000× *g* for 20 min at 4 °C. The resulting supernatant was carefully collected and diluted with mass spectrometry-grade water to obtain a solution containing 53% methanol. This solution was once again centrifuged at 15,000× *g* for 20 min at 4 °C, and the resulting supernatant was collected. To validate the methodology, a quality control (QC) sample was prepared by mixing 20 µL of each sample. This QC sample was then analyzed using ultra high performance liquid chromatography (UHPLC-MS/MS).

#### 4.4.2. Chromatography and Mass Spectrometry Conditions

UHPLC-MS/MS analysis for metabolomics was performed using a Vanquish UHPLC system (Thermo Fisher, Waltham, MA, USA) in conjunction with an Orbitrap Q ExactiveTM HF mass spectrometer. The samples were separated on a Hypesil Gold Column (100 × 2.1 mm, 1.9 µm) at a flow rate of 0.2 mL/min. The positive polarity mode mobile phase consisted of (A) 0.1% formic acid in water, and (B) methanol. The negative polarity mode mobile phase consisted of (A) 5 mM ammonium acetate, pH 9.0, and (B) methanol. The solvent gradient was set as follows: 2% B, 1.5 min; 2–100% B, 12.0 min; 100% B, 14.0 min; 100–2% B, 14.1 min; 2% B, 17 min. The Q ExactiveTM HF mass spectrometer was operated in positive/negative polarity mode with a spray voltage of 3.5 kV, a capillary temperature of 320 °C, a gas flow rate in the sheath of 35 psi, an auxiliary gas flow rate of 10 L/min, an S-lens RF level of 60, and an auxiliary gas heater temperature of 350 °C.

#### 4.4.3. Analysis of Metabolomics Results

The UHPLC-MS/MS raw data files were imported into Compound Discoverer 3.1 (CD3.1, Thermo Fisher) for peak alignment, extraction, and relative area quantification. The peak intensities were also normalized to the total spectral intensity to enable comparison of data across different levels. Sample information, retention times, peak intensities, and other relevant data were gathered using Excel software (version 2010).

Principal component analysis (PCA) and orthogonal partial least squares discriminant analysis (OPLS-DA) were conducted using SIMCA-14.1 software to distinguish among the various groups. The selection of significant variables for projection values was based on VIP > 1.0, and independent t-tests were applied at a significance level of *p* < 0.05 to identify differential metabolites. Pathway analysis of the identified metabolites was performed using MetaboAnalyst 5.0.

### 4.5. Network Pharmacology Analysis

#### 4.5.1. Drug Composition and Disease Target Screening

We applied initial screening conditions of oral bioavailability (OB) ≥ 30% and drug-likeness (DL) ≥ 0.18 for the Traditional Chinese Medicine Systems Pharmacology Database and Analysis Platform (TCMSP) (http://lsp.nwu.edu.cn/tcmsp.php, accessed on 7 July 2023) [39], and complemented XYS with UPLC-MS to identify key components with significant pharmacological activities that were not screened in TCMSP. To predict the targets of XYS components, we utilized TCMSP and SwissTargetPrediction (http://www.swisstargetprediction.ch/, accessed on 7 July 2023) [40], and transformed the results into the Uniprote database (https://www.uniprot.org/, accessed on 7 July 2023) [41]. We conducted searches on Genecard (http://genecards.org/, accessed on 9 July 2023) [42], the Therapeutic Target Database (TTD) (http://bidd.nus.edu.sg/group/cjttd/, accessed on 9 July 2023) [43], Drugbank (https://go.drugbank.com/, accessed on 9 July 2023) [44], and the Online Mendelian Inheritance in Man (OMIM) (https://omim.org/, accessed on 9 July 2023) database [45] using “depression” and “inflammation” keywords for disease targets, and selected genes with a relevance score >3 as targets in Gene-card. We collated the results using Microsoft Excel (2010) and obtained the intersections of XYS component targets and disease targets using Venny 2.1.

#### 4.5.2. PPI Network Construction

To create protein–protein interaction (PPI) networks of intersection targets, the latest version of the STRING database was used [46], and only networks with interaction scores above a threshold of 0.04 were selected. The resulting PPI networks were visualized using Cytoscape (version 3.8.2) [47], and a network analysis plug-in was employed to calculate the degree value for each node, which represents the degree of association with other nodes (targets with high degree values were considered more important in the network). Topological analysis was performed using a CytoNCA plug-in to identify core genes with degree values ≥ 2 times the median, mean centrality, and mean tightness [48].

#### 4.5.3. GO and KEGG Enrichment Analyses

The Metascape tool was used to perform enrichment analysis for Gene Ontology (GO) and the Kyoto Encyclopedia of Genes and Genomes (KEGG) [49]. Enrichment results with a significance level of *p* < 0.05 were considered significant. The online tool (http://www.bioinformatics.com.cn, accessed on 10 July 2023) was used to visualize the KEGG and GO results.

#### 4.5.4. Molecular Docking

Molecular docking was conducted to confirm the binding affinity of the selected core components to key targets. The target protein’s 3D crystal structure was obtained from the RCSB database (http://www.rcsb.org/pdb, accessed on 29 July 2023) [50], while the 3D structure of the compound ligand was retrieved from PubChem [51]. We utilized OpenBabel 3.1.1 (http://openbabel.org/wiki/Main_Page, accessed on 29 July 2023) to transform the format of the ligand-receptor file into pdb format [52]. AutoDock 4.2 was then applied for essential procedures including receptor dehydration and hydrogenation, energy minimization of the chemosynthetic allocator, ligand atom type assignment, and Gasteiger charge calculations [53]. Finally, we saved the file in pdbqt format. Thereafter, we performed molecular docking using AutoDock Vina to screen for active ingredients with superior binding activity to the target using the docking score Affinity [54]. The scores for each combination were tallied, and the top four positions of activity were visualized using PyMol. Finally, we constructed a heat map of ligand-receptor binding energy with Origin.

### 4.6. Integrating Analysis

The core proteins were identified by searching for the relevant targets of the differential metabolites in the Human Metabolome Database (HMDB) and intersecting them with the Hub targets in the Network Pharmacology Analysis, thereby obtaining a list of potential proteins. The accuracy of the screened core proteins was verified through molecular docking and experimental analysis.

### 4.7. Enzyme Linked Immunosorbent Assay (ELISA)

The serums of six mice were selected from each group. The experiments were performed using the ELISA kit method. After color development, absorbance was measured at 450 nm using a fluorescence analyzer (THERMO USA), and protein levels of IL-6, TNF-α, and IL-1β were then calculated.

### 4.8. Immunohistochemistry

The spleen tissues were fixed in 4% paraformaldehyde for three days, and dehydrated. The sections were then dewaxed with xylene and then hydrated by gradient concentration ethanol. Tissue antigens were repaired using the high-pressure method, and the PV-9000 two-step immunohistochemical assay kit was utilized. An endogenous peroxidase blocker was added to the tissue sections, which were then incubated at room temperature for 10 min. The corresponding primary antibodies, PPARG (16643-1-AP, 1:500 dilution, Proteintech, Rockford, IL, USA) and VEGFA (DF7470, 1:500 dilution, Affinity BioSciences, Newark, NJ, USA), were added to the sections and incubated overnight at 4 °C. The primary antibody was recovered the next day, and the reaction enhancement solution was added to the tissue sections and incubated at room temperature for 20 min. Subsequently, the sections were incubated with the appropriate amount of universal anti-mouse/rabbit secondary antibody at 37 °C for 30 min, and DAB chromogenic solution was used for color development. Finally, the sections were stained with hematoxylin and sequentially dehydrated with gradient alcohol, dehydrated with xylene, and sealed.

### 4.9. Real Time Quantitative PCR

Total RNA was extracted from the mouse spleen tissue using the Tissue RNA Purification Kit from EZBioscience (Roseville, CA, USA) in accordance with the protocol. Subsequently, cDNA was synthesized using the Reverse Transcription Color Reverse Transcription Kit from EZBioscience (USA). The RT-qPCR experiment was conducted using the 2xColor SYBR Green qPCR Master Mix (ROX2 plus) kit from EZBioscience (USA), and the real-time fluorescent quantitative PCR system (BioRad, CFX96, Hercules, CA, USA) was utilized to run the PCR. The primer sequences were provided by Sangon Biotech (Shanghai, China). The forward primer of VEGFA was GGCCTCCGAAACCATGAACT, and the reverse primer of VEGFA was GTCCACCAGGGTCTCAATCG. The forward primer of PPARG was CCAGAAGCCTG-CATTTCTGC, and the reverse primer of PPARG was GTGTCAACCATGGTCATTTCGTT. The 2^−ΔΔCt^ method was used to calculate the relative expression of genes, where Ct denotes the threshold cycle.

### 4.10. Western Blot

The spleen tissues were washed three times with pre-cooled PBS. RIPA (P0013B, Beyotime, Shanghai, China), a phosphatase inhibitor, and a protease inhibitor (100:1:1) were added to the tissues, homogenized in a low-temperature homogenizer, lysed on ice for 30 min, and centrifuged at 4 °C at 13,000 rpm, after which the supernatant was removed. The protein content was determined using the BCA Protein Assay Kit (P0010-1, Beyotime, Shanghai, China). The total protein was run on tris-glycine 10% SDS-page gels (Biosharp, Hefei, China) at 80 V for 0.5 h and 110 V for 1 h. The gels were transferred onto polyethylene difluoride (PVDF) membranes (Bio-Rad, USA) and blocked in 5% skimmed milk for 2 h. The PVDF membranes were washed three times with TBST and incubated overnight at 4 °C with primary antibodies VEGFA (1:1000, DF7470, Affinity, USA), PPARG (1:4000, 16643-1-AP, Proteintech, USA), and GAPDH (1:5000, 60004-1-IG, Proteintech, USA). The corresponding secondary antibodies were incubated at room temperature for 2 h and then washed three times with TBST. A fully automated chemiluminescence image analysis system (BioRad, Hercules, CA, USA) was used to capture images of the proteins on the membranes, and ImageJ software (v1.8.0) was used to analyze the expression of each group of proteins.

### 4.11. Statistical Analysis

All data were analyzed for statistical significance using SPSS 20.0 software (SPSS, Chicago, IL, USA) and visualized through GraphPad Prism 8.0 software (San Diego, CA, USA). The results were presented as the mean ± standard deviation. Statistical differences were evaluated using one-way ANOVA analysis. A *p*-value < 0.05 was considered statistically significant.

## 5. Conclusions

In this study, we used UHPLC-MS/MS metabolomics combined with network pharmacology to analyze the modulation of depressive inflammatory responses by XYS. We systematically described the findings, including the identification of 21 differential metabolites and four key metabolic pathways in spleen metabolomics. VEGFA and PPARG have been identified as potential key targets involved in the antidepressant inflammatory response of XYS. To the best of our knowledge, this study provides the first evidence that XYS has the ability to modulate multiple biological pathways and induce metabolic changes in order to attenuate the inflammatory response associated with depression. This is achieved through targeting specific mechanisms and promoting appropriate metabolism. This study contributes to our understanding of the theory of “prevention of disease” in Traditional Chinese Medicine from the perspective of modern medical science, and promotes the application of TCM theories in modern clinical practice.

## Figures and Tables

**Figure 1 pharmaceuticals-16-01607-f001:**
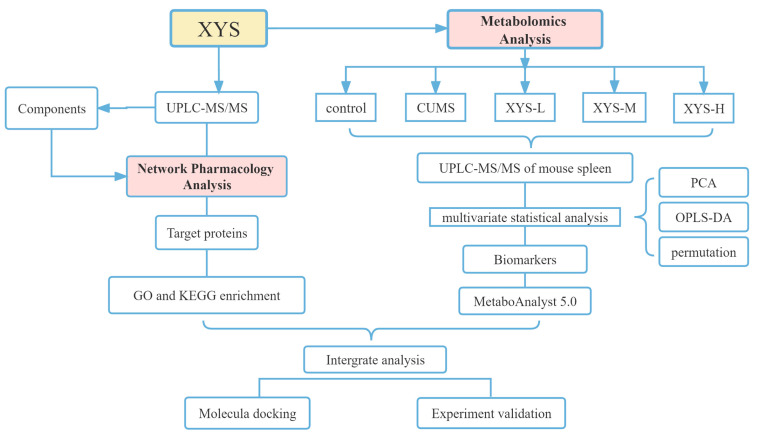
Experimental flowchart.

**Figure 2 pharmaceuticals-16-01607-f002:**
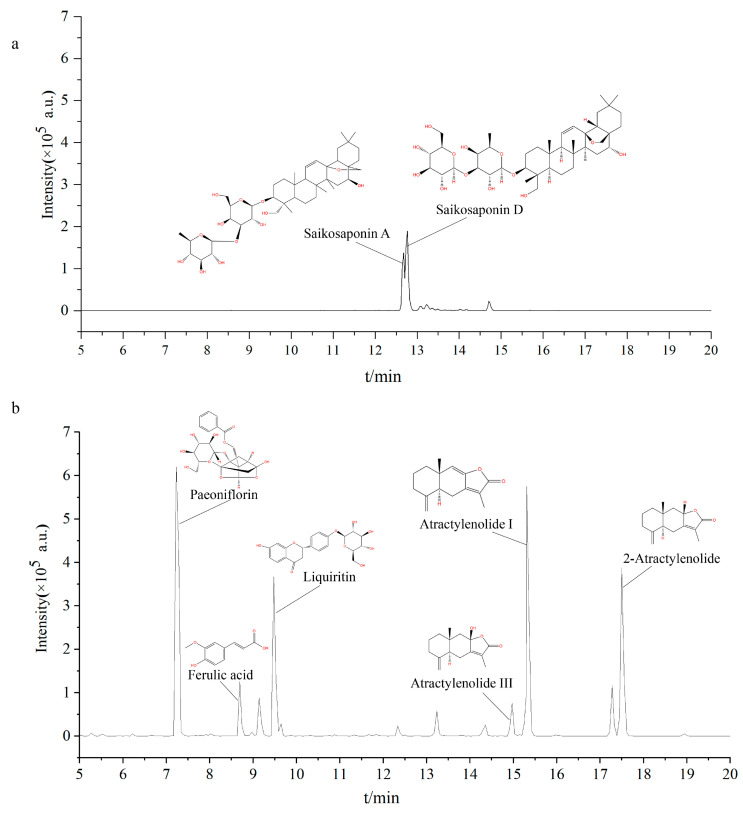
Identification of chemical components of XYS. Total Ion Chromatogram (TIC) in positive (**a**) and negative (**b**).

**Figure 3 pharmaceuticals-16-01607-f003:**
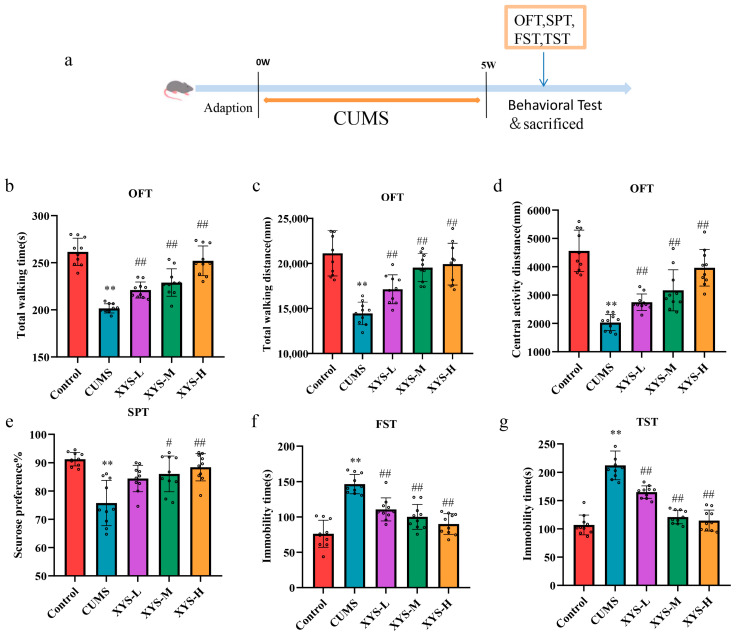
Effect of Xiao-Yao-San on depression-like behaviors in chronic unpredictable mild stress mice. (**a**) Mouse moulding time flow chart. (**b**–**d**) Open Field Test (OFT). (**e**) Sucrose Preference Test (SPT). (**f**) Forced Swimming Test (FST). (**g**) Tail Suspension Test (TST). Data were presented mean ± SD, *n* = 10 pre group. ** *p* < 0.01 compared with the control group, ^#^ *p* < 0.05, ^##^ *p* < 0.01, compared with the CUMS group.

**Figure 4 pharmaceuticals-16-01607-f004:**
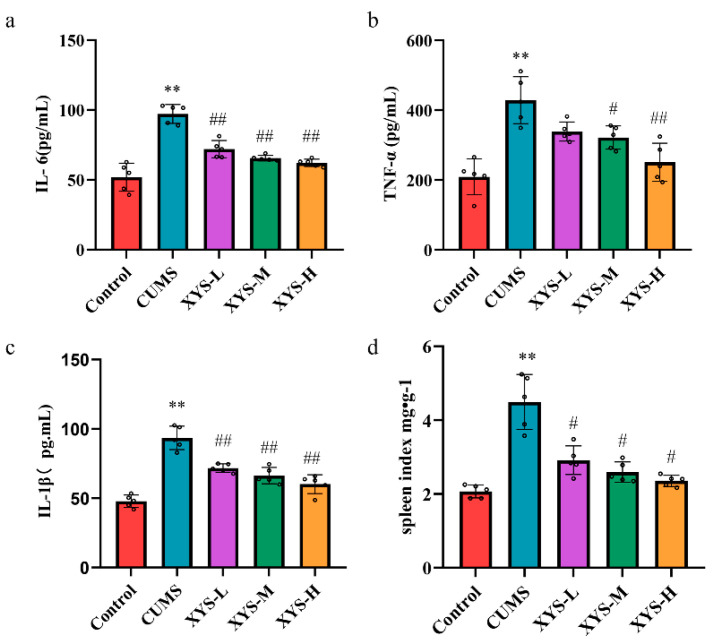
Effect of XYS on the inflammatory response in the serum of CUMS mice. (**a**) IL-6 level. (**b**) TNF-α level. (**c**) IL-1β level. (**d**) Spleen index. Data were presented mean ± SD, *n* = 6 pre group. ** *p* < 0.01 compared with the control group, ^#^ *p* < 0.05, ^##^ *p* < 0.01, compared with the CUMS group.

**Figure 5 pharmaceuticals-16-01607-f005:**
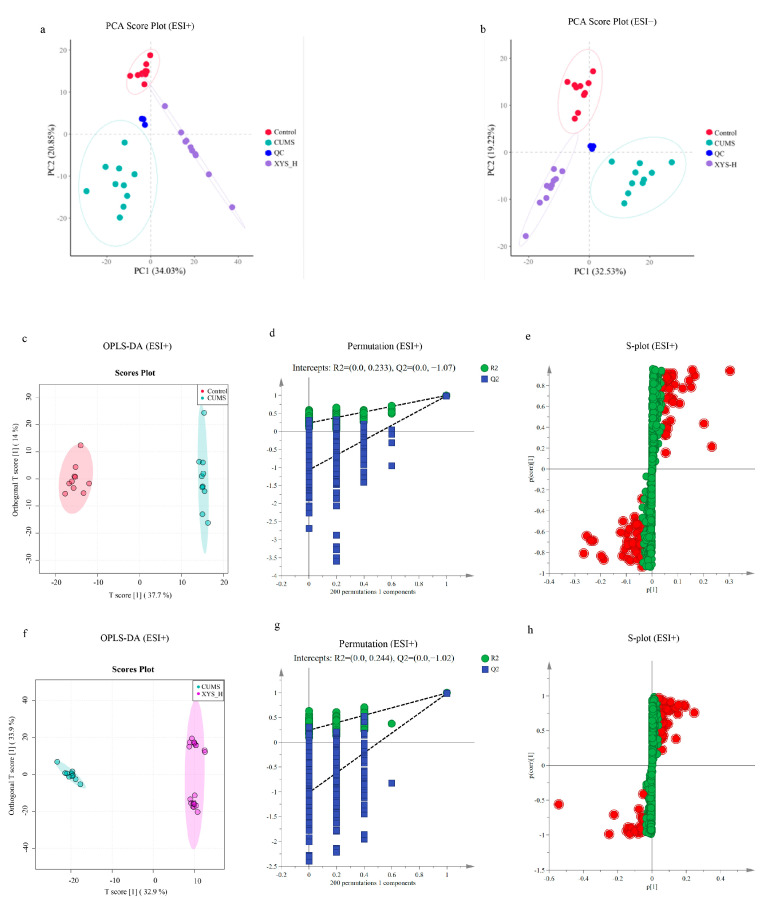
Multivariate statistical analysis of metabolic characters of mouse spleen samples. PCA score plot of metabolomic analysis in the ESI+ model (**a**) and ESI− model (**b**). The OPLS-DA score graph (**c**), permutation (**d**) and S-plot (**e**) of the control vs. CUMS groups as well as the plot of OPLS-DA (**f**) permutation (**g**) and S-plot (**h**) of CUMS vs. XYS-H for positive ion mode. In the S-plot, the red markers indicate metabolites with VIP values ≥ 1, while the green markers represent metabolites with VIP values < 1.

**Figure 6 pharmaceuticals-16-01607-f006:**
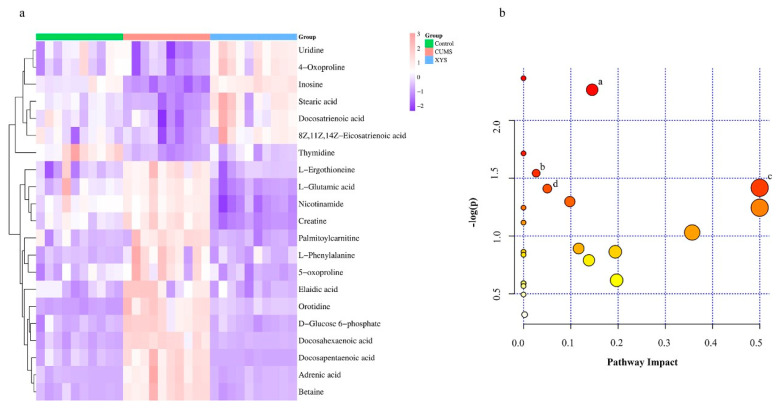
Heatmap visualization of the 21 potential metabolites expressed in mouse spleens (**a**). Pathway analysis of the specific metabolites (**b**): a. Pyrimidine metabolism, b. Glutathione metabolism, c. Phenylalanine, tyrosine and tryptophan biosynthesis, d. Glycine, serine, and threonine metabolism. In the bubble plot, larger and darker bubbles represent a stronger correlation between the metabolic pathway and the differential metabolite.

**Figure 7 pharmaceuticals-16-01607-f007:**
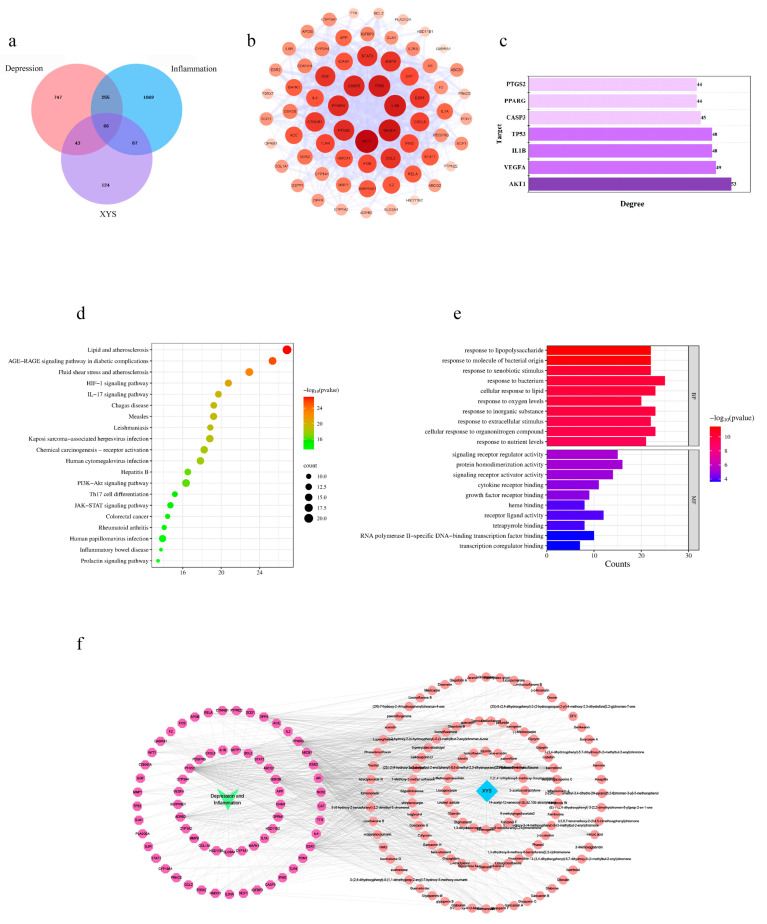
Network pharmacological analysis of XYS in the treatment of depression inflammation. (**a**) Venn diagram of overlapping targets. (**b**) Intersection target PPI viewable. The higher the degree of the node, the darker the node will be. The thickness of the edges is related to the combined score. (**c**) Top 10 key targets in terms of degree. (**d**) The KEGG pathway enrichment analysis of the potential targets. (**e**) GO terms enrichment analysis. (**f**) The disease-ingredients-target network map, with disease names labeled in green, intersecting targets labeled in purple, drugs labeled in light blue, and chemical components labeled in pink.

**Figure 8 pharmaceuticals-16-01607-f008:**
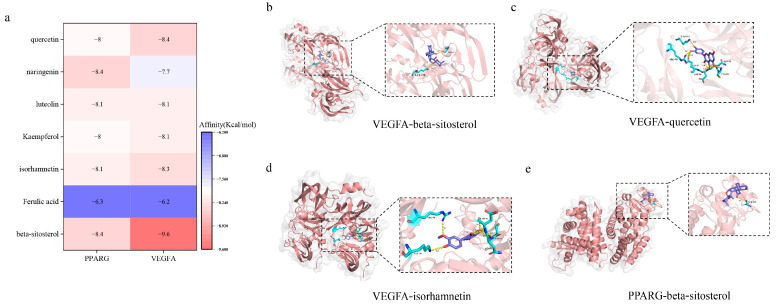
Molecular docking results. (**a**) Heat map of ligand-receptor binding energy. (**b**–**e**) Molecular docking pattern diagram.

**Figure 9 pharmaceuticals-16-01607-f009:**
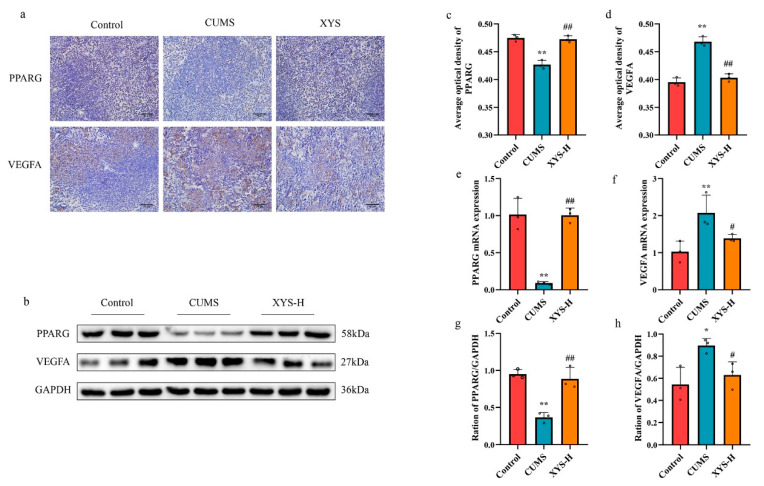
Immunohistochemistry (magnification 400×, *n* = 3) (**a**,**c**,**d**), RT-qPCR (**e**,**f**) and Western Blot (**b**,**g**,**h**) analysis of key targets. Data are presented as mean ± SD, *n* = 3 pre group. * *p* < 0.05, ** *p* < 0.01 compared with the control group, ^#^ *p* < 0.05, ^##^ *p* < 0.01, compared with the CUMS group.

**Table 1 pharmaceuticals-16-01607-t001:** Chemical composition of XYS aqueous solution.

NO.	Compound Name	Formula	Retention Time	*m*/*z* (Da)	OB
1	Paeoniflorin	C_23_H_28_O_11_	6.3	481.17	53.87%
2	Liquiritin	C_21_H_22_O_9_	8.9	419.13	29.23%
3	Ferulic acid	C_10_H_10_O_4_	7.9	195.07	39.56%
4	Atractylenolide I	C_15_H_18_O_2_	15.3	231.14	37.37%
5	2-Atractylenolide	C_15_H_20_O_2_	14.9	233.15	47.50%
6	Atractylenolide III	C_15_H_20_O_3_	15.3	249.15	31.15%
7	Saikosaponin A	C_42_H_68_O_13_	12.8	779.46	32.39%
8	Saikosaponin D	C_42_H_68_O_13_	14.8	779.46	34.39%

OB: Oral bioavailability.

**Table 2 pharmaceuticals-16-01607-t002:** Potential differential metabolites.

NO.	Name	Formula	ESI Mode	Retention Time	*m*/*z*	CUMS vs. Control		XYS vs. CUMS	
						VIP	Trend	VIP	Trend
1	Nicotinamide	C_6_H_6_N_2_O	[M+H]+	1.978	123.055	2.9031	↓	3.8747	↑
2	Betaine	C_5_H_12_NO_2_	[M+H]+	1.331	118.0861	4.28352	↑	3.35698	↓
3	L-Glutamic acid	C_5_H_9_NO_4_	[M+H]+	1.324	148.0603	1.73996	↓	2.68374	↑
4	Creatine	C_4_H_9_N_3_O_2_	[M+H]+	1.376	132.0766	2.38651	↓	2.399	↑
5	5-oxoproline	C_5_H_7_NO_3_	[M+H]+	1.321	130.0499	1.7047	↓	2.15806	↑
6	Inosine	C_10_H_12_N_4_O_5_	[M+H]+	3.446	269.0882	1.85151	↑	1.57746	↓
7	Palmitoylcarnitine	C_23_H_46_NO_4_	[M+H]+	7.901	400.3421	5.02181	↓	1.43203	↑
8	L-Phenylalanine	C_9_H_11_NO_2_	[M+H]+	5.118	166.0859	1.32897	↓	1.40082	↑
9	Ergothioneine	C_9_H_15_N_3_O_2_S	[M+H]+	1.397	230.0955	2.03896	↑	1.29377	↓
10	Docosahexaenoic acid	C_22_H_32_O_2_	[M-H]−	9.767	327.2324	3.33233	↑	6.97933	↓
11	Adrenic acid	C_22_H_36_O_2_	[M-H]−	10.518	331.2638	6.92823	↑	4.99596	↓
12	Docosapentaenoic acid	C_22_H_34_O_2_	[M-H]−	10.054	329.2479	2.73121	↑	4.28855	↓
13	8Z,11Z,14Z- Eicosatrienoic acid	C_20_H_34_O_2_	[M-H]−	10.371	305.2482	1.78561	↑	2.09322	↓
14	D-Glucose6- phosphate	C_6_H_13_O_9_P	[M-H]−	1.228	259.0227	1.62244	↑	1.40387	↓
15	4-Oxoproline	C_5_H_7_NO_3_	[M-H]−	2.245	128.0354	1.47797	↓	1.34707	↑
16	Stearic acid	C_18_H_36_O_2_	[M-H]−	11.152	283.2641	1.47313	↑	1.28179	↓
17	Uridine	C_9_H_12_N_2_O_6_	[M-H]−	2.285	243.0623	1.85047	↑	1.15761	↓
18	Elaidic acid	C_18_H_34_O_2_	[M-H]−	10.601	281.2483	2.30575	↑	1.85233	↓
19	Thymidine	C_10_H_14_N_2_O_5_	[M-H]−	4.878	241.0832	1.98812	↑	1.37354	↓
20	Docosatrienoic acid	C_22_H_38_O_2_	[M-H]−	11.038	333.28	1.85929	↑	1.3362	↓
21	Orotidine	C_10_H_12_N_2_O_8_	[M-H]−	1.347	333.0593	1.6326	↑	1.29154	↓

**Table 3 pharmaceuticals-16-01607-t003:** Enrichment analysis of potential metabolic pathways.

NO.	Pathway Name	Total	Expected	Hits	*p*-Value	Impact
1	Pyrimidine metabolism	39	0.37742	3	0.005439	0.14535
2	Glutathione metabolism	28	0.27097	2	0.028584	0.02675
3	Phenylalanine, tyrosine and tryptophan biosynthesis	4	0.03871	1	0.038188	0.5
4	Glycine, serine and threonine metabolism	33	0.31935	2	0.038829	0.05034

**Table 4 pharmaceuticals-16-01607-t004:** Hub genes obtained by topological analysis.

NO.	Gene Name	Degree	Betweenness	Closeness
1	*AKT1*	53	203.9459	0.833333
2	*VEGFA*	49	108.068	0.792683
3	*IL1β*	48	201.6406	0.792683
4	*TP53*	48	159.9084	0.783133
5	*CASP3*	45	81.21995	0.755814
6	*PPARG*	44	187.2315	0.747126
7	*PTGS2*	44	88.78311	0.747126

**Table 5 pharmaceuticals-16-01607-t005:** Hub ingredients obtained by topological analysis.

NO.	Name	Degree	Betweenness	Closeness
1	Quercetin	83	4856.966	0.521994
2	Kaempferol	55	1488.816	0.459948
3	Naringenin	27	747.0437	0.450633
4	Beta-sitosterol	25	370.6822	0.439506
5	Luteolin	22	525.4311	0.469657
6	Isorhamnetin	21	229.8738	0.44389
7	Ferulic acid	19	957.7249	0.462338

## Data Availability

Data is contained within the article and Appendix A.

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
