# Peer review of "Integrating Metabolomics and Network Pharmacology to Explore the Mechanism of Xiao-Yao-San in the Treatment of Inflammatory Response in CUMS Mice"

_pharmaceuticals, 2023, doi:10.3390/ph16111607_

Round 1

Reviewer 1 Report

Comments and Suggestions for Authors

The manuscript written by Zhang et al presents interesting research on integrating metabolomics and network pharmacology in determining the mechanism of XYS formulation in CUMS mice. However, the study outcome does not achieve the target of determining the mechanism.

Authors should follow a rule that before using any abbreviation in the abstract, text, or table, they should expand them, I could not find this rule implementation. Also, only those words should be abbreviated that will be used at least twice in that section, otherwise, they must use full form.

The author needs to give more details about XYS formulation, as they say, that it is an established formulation for depression, it needs to be described in more detail in the materials section. Also, include the source of this formulation with the Lot/Batch number.

Further, I found in section 4.3, I found that they added some ratios, from where did they get this information, what is the reference, and any toxicity study for these extracts?

Remove the reference from the heading of 4.2.1, and include it within the section.

I could not find the experimental group classification section. How many times, the selected samples were administered to each of the animals? Why did they select male mice for the study? Any significance of selection?

In the results section, I found that there are three doses of the formulation used, and how they were selected.

There are several studies that reported the study, it is very confusing the understand their sequence, authors can help readers with the help of a flow chart that should also include different groups, their treatment, and number and show, what sequence were followed in the experimental setup.

Use consistently only one group name throughout the text, somewhere, it is mentioned as blank, somewhere as control.

Overall, the resolution of the figures should be improved.

Figure 5a needs more clarification.

Abbreviations should be expanded in the table legend.

Comments on the Quality of English Language

Acceptable language.

Author Response

On behalf of all the co-authors, we would like to express our sincere appreciation to the reviewers for providing valuable comments on our manuscript. We have thoroughly reviewed and addressed each comment in order to address the concerns raised by the reviewers. In the following paragraphs, we provide detailed responses to each point raised by the reviewers. We hope that the revised version meets the standards required for publication. Thank you once again for your invaluable feedback.

1. The manuscript written by Zhang et al presents interesting research on integrating metabolomics and network pharmacology in determining the mechanism of XYS formulation in CUMS mice. However, the study outcome does not achieve the target of determining the mechanism.

Answer: The research team is using this study as an initial exploration into the potential improvement of the inflammatory state under depression by XYS. Further investigations will be conducted through in-depth experimental research, focusing on the key components and targets identified in this study, to elucidate the underlying mechanisms.

2. Authors should follow a rule that before using any abbreviation in the abstract, text, or table, they should expand them, I could not find this rule implementation. Also, only those words should be abbreviated that will be used at least twice in that section, otherwise, they must use full form.

Answer: Thank you for the suggestion and that is very important. We have diligently followed the rule by expanding the full term for any terminology mentioned for the first time in the manuscript.

3. The author needs to give more details about XYS formulation, as they say, that it is an established formulation for depression, it needs to be described in more detail in the materials section. Also, include the source of this formulation with the Lot/Batch number.

Answer: Thank you for the suggestion. We are completely agree with your comments. The source and the Lot/Batch number of the drugs used in the Xiao-yao-san formulation have been described in Section 4.1 of the manuscript as recommended.

4. Further, I found in section 4.3, I found that they added some ratios, from where did they get this information, what is the reference, and any toxicity study for these extracts?

Answer: Thanks for this assessment. The proportions of Xiaoyao San mentioned in Section 4.3 were determined based on previous research conducted by our research group and existing experimental studies as reference. We sincerely apologize for not clearly describing this in the text, and have now included the relevant references in the paper.

There are no toxic components in the composition of Xiaoyao San. Moreover, in previous experiments, the use of medication proportions did not result in any toxic side effects. In this study, the highest dosage used was four times the adult clinical dose, and no cases of mouse mortality were recorded during the experiments. Additionally, the medication at four times the concentration showed significant therapeutic effects.

5. Remove the reference from the heading of 4.2.1, and include it within the section.

Answer: Thank you for your kind remind. We apologize for this mistake. We have removed the citations from the headings and placed the references at the end of that section.

6. I could not find the experimental group classification section. How many times, the selected samples were administered to each of the animals? Why did they select male mice for the study? Any significance of selection?

Answer: Thank you for your comments. In our experiment, the animal experimental classification has already been described in Section 4.2 of the manuscript as follows: Fifty mice were randomly assigned to five groups, namely, the control group, chronic unpredictable mild stress (CUMS) group, CUMS + low-dose XYS (XYS-L, 1.4g/kg) group, CUMS + medium-dose XYS (XYS-M, 2.8g/kg) group, and CUMS + high-dose XYS (XYS-H, 5.6g/kg) group.

We sincerely apologize for the oversight in not clearly indicating the duration of drug administration. The revised statement is as follows: "CUMS+ low-dose XYS (XYS-L, 1.4g/kg/d), CUMS+ medium-dose XYS (XYS-M, 2.8g/kg/d), and CUMS+ high-dose XYS (XYS-H, 5.6g/kg/d) were continuously administered for a duration of 5 weeks." We have made the necessary revisions in the manuscript.

As the main focus of this study is the measurement of inflammatory factors, which are significantly affected by hormonal levels, we intentionally excluded female mice due to their tendency to experience physiological fluctuations in hormone levels. Instead, male mice were selected for the experiment.

7. In the results section, I found that there are three doses of the formulation used, and how they were selected.

Answer: Thank you for the suggestion and that is very important. We apologize for our unclear description. In the revised manuscript, we had provided more detailed description of the dose of XYS in 4.2 Preparation of Drugs as following “The dose of XYS in the XYS-L group mice was calculated by converting it based on the standard human clinical dosage and the administration dosage conversion coefficient from human to mouse (The administration dosage in the XYS-L group = 9.1× human drug dosage / 60 kg) (1.4 g/kg/d). Subsequently, the medium dose (2.8 g/kg/d) and high dose (5.6 g/kg/d) were considered as 2-fold and 4-fold of the initial dose, respectively). The control and model groups were administered an equivalent volume of saline water by intragastric gavage. Additionally, the mice received drug treatment continuously during the 5-week CUMS procedure.”

8. There are several studies that reported the study, it is very confusing the understand their sequence, authors can help readers with the help of a flow chart that should also include different groups, their treatment, and number and show, what sequence were followed in the experimental setup.

Answer: Thank you very much for providing the valuable suggestions. We are completely agreed with your comments. In the revised manuscript, we had added figure of experimental protocols.

9. Use consistently only one group name throughout the text, somewhere, it is mentioned as blank, somewhere as control.

Answer: Thank you for your kind remind. We apologize for this mistake. In the revised paper, we have renamed the blank group as the control group.

10. Overall, the resolution of the figures should be improved.

Answer: Thank you for your comments. The images we have submitted have been edited and uploaded in the required resolution (300 dpi) specified by the journal.

11. Figure 5a needs more clarification.

Answer: Figure 5a displays a heat map illustrating the differential metabolite concentrations. In this representation, the intensity of the color on the grid corresponds to the concentration level of the differential metabolites. Dark purple grid colors indicate lower concentrations, whereas dark red grid colors indicate higher concentrations of the differential metabolites.

12. Abbreviations should be expanded in the table legend.

Answer: We are completely agree with your comments. Thank you very much for giving us this idea. We've added a legend for the abbreviations in the manuscript.

Reviewer 2 Report

Comments and Suggestions for Authors

the article need some English editing

  1. some references is to old since 2008, 2009 you need to modify it 
  2. the innovation of the work is moderate as many studies  was done on application on a certain drug on prevention of diseas in Traditional Medicine 
  3. in figure 4a we must remove word (and) 
  4. I would like the author to send the original file for figure 8a

Comments on the Quality of English Language

the article need English editing

Author Response

在此,我谨代表所有合著者,衷心感谢审稿人对我们稿件的宝贵意见。我们仔细研究和考虑了每条意见,并相应地进行了适当的修改。

1.有些参考资料从2008年开始太旧了,2009年需要修改

答:谢谢你的建议,这非常重要。我们更新了 2009 年及之前的参考文献引用,将其替换为最新的参考文献。(我们用红色做了注释,以指示我们替换参考文献的位置。

2.在传统医学中对某种药物预防疾病的应用进行了许多研究,因此这项工作的创新性适中

答:临床研究发现抑郁症患者的炎症因子失调。然而,炎症因子失调是各种炎症相关疾病发生发展的关键因素,包括其与癌症发展的关联以及促进不同阶段的肿瘤形成 (1)。本研究旨在遵循中医"防病、防变"理论,改善炎症因子失调所致下游炎症相关疾病的发生与进展。本研究拓展了小瑶散的应用思路和方向。

  1. Greten FR, Grivennikov SI.炎症和癌症:触发因素、机制和后果。免疫。2019;51(1):27-41.

3. 在图 4a 中,我们必须删除单词(和)

答:感谢您的评估。在图 4a 中,没有出现“和”一词。但是,如果您在图例中引用"和"一词,则其用法符合期刊的文献指南,并且是允许的。如果您需要任何进一步的校对或编辑帮助,请向我提供具体的句子,我很乐意修改。

4. 我希望作者发送图 8a 的原始文件

答:由于期刊系统的限制,作者只能提交单个Word或PDF文件,因此我将图8a的原始图像放在修改后的稿件末尾。(原文件夹中标有"图8a"的图片是指稿件中显示的图片,另外两张图片来自平行实验。

Reviewer 3 Report

Comments and Suggestions for Authors

Dear Authors: Depression is a common clinical condition, with an even worse incidence of it after the pandemic Covid-19 occurred, Traditional Chinese Medicine can play an important treating effect. Xiaoyao San is a basic formula of TCM which is utilized in the clinic to treat depression and many kinds of psychological disorders. You selected a significant topic. you integrate metabolomics and network pharmacology to explore the mechanism of Xiaoyao San via the treatment of inflammatory response in mice which you established as an animal model with CUMS (Chronic unpredictable mild stress). The positive result is collected from your scientific, comprehensive and objective observation procedure. The result is factual and credible. Your research will contribute to a feasible improvement in the clinical application of Xiaoyao San. So I believe this is a very successful design and significant research. Congratulation. 

Author Response

Thank you very much for your review and support. We will continue to devote ourselves to in-depth research on the possible mechanisms of Xiao-yao-san in the prevention and treatment of depression and related inflammatory disorders. As a traditional Chinese medicine formula, XYS has a long history and extensive experience in clinical application. However, in the context of modern medicine, we aim to reveal its pharmacological effects and specific mechanisms through more basic research. We are committed to further exploring the mechanism of action of the formula in the treatment of depression and related inflammatory disorders, with the hope of providing a more solid scientific foundation and support for its clinical application through specific and systematic research work. We eagerly look forward to future research results and strive to further improve the health of patients.

Reviewer 4 Report

Comments and Suggestions for Authors

The manuscript "Integrating Metabolomics and Network pharmacology to explore the Mechanism of Xiao-Yao-San in Treatment of Inflammatory Response in CUMS" presents the proposal of an unprecedented work on the modulation of depressive inflammatory responses by XYS. The identification of differential metabolites and metabolic pathways involved in spleen metabolomics have been very well described.

I believe the paper will make a great contribution to the journal. Below are my comments:

- What could be the possible interferences or limitations in the efficacy of the treatment of depression using Xiao-Yao-San (XYS)?

- Describe future steps to validate the use of Xiao-Yao-San (XYS) in humans.

- Review literature citations that mention different uses of Xiao-Yao-San (XYS) in the prevention/treatment of other diseases.

- Based on the results found, discuss possible interactions of the simultaneous use of Xiao-Yao-San (XYS) with conventional medications used in the treatment of depression.

Author Response

On behalf of all the co-authors, we would like to express our heartfelt appreciation to the reviewers for providing their valuable comments on our paper. We have thoroughly reviewed and addressed each of the comments with utmost care. I hope that the reviewers will acknowledge and recognize our responses.

1. What could be the possible interferences or limitations in the efficacy of the treatment of depression using Xiao-Yao-San (XYS)?

Answer: We are appreciative of this assessment. Given that Xiao-Yao-San (XYS) is a complex formulation comprised of multiple medicinal components, each herb may contribute to varied therapeutic effects due to differences in origin and preparation techniques. For that reason, we've chosen to use granules, which offer higher controllability, to maintain stability in its basic active ingredients and overall efficacy.

2. Describe future steps to validate the use of Xiao-Yao-San (XYS) in humans.

Answer: Thanks for your insightful comment. Xiao-Yao-San (XYS) is a formulation comprising multiple active compounds, and the precise components responsible for its effects have yet to be fully elucidated. In future basic research, it would be beneficial to concentrate on investigating the key components or combinations of components of XYS. This approach aims to facilitate the development of drug standards, batch production, and to enhance the standardization and safety of medication. Additionally, such research can help improve patient compliance and acceptance of medication.

3. Review literature citations that mention different uses of Xiao-Yao-San (XYS) in the prevention/treatment of other diseases.

Answer: Thanks for this assessment. XYS has the beneficial effect of regulating liver and spleen functions. Based on the Chinese medicine theory of "treating different diseases with the same treatment", XYS can be used in the treatment of various conditions related to liver depression, spleen deficiency, and blood deficiency. These conditions include anxiety disorders, insomnia, indigestion, menstrual irregularities, and more. XYS has been widely used in clinical practice and has shown good results. However, further research is needed to fully understand its precise pharmacological effects and underlying mechanisms. Therefore, our focus remains on exploring the indications for the use of XYS and providing more scientific evidence to support its rational application, with the goal of achieving better clinical outcomes.

4. Based on the results found, discuss possible interactions of the simultaneous use of Xiao-Yao-San (XYS) with conventional medications used in the treatment of depression.

Answer: Thank you for your comments. XYS is a traditional herbal formula that has been used for nearly a thousand years and has unique advantages in the treatment of depression. Compared to classic antidepressant medications, XYS has fewer adverse effects and is relatively safe, allowing for long-term use. Research has shown that combining XYS with selective serotonin reuptake inhibitors (SSRIs) and other conventional antidepressant medications can not only reduce the adverse effects of SSRIs but also lead to better antidepressant efficacy. XYS can regulate liver and spleen functions, alleviating symptoms such as emotional tension, insomnia, and indigestion, thereby improving depressive states. Additionally, XYS has the ability to balance Qi and blood, and regulate the immune system and neuroendocrine system, further enhancing its antidepressant effects. The combined therapy of XYS and SSRIs has been found to result in sustained improvement in depressive states and extended duration of antidepressant effects. Compared to using SSRIs alone, the combination therapy not only strengthens the antidepressant effects but also mitigates the side effects of SSRIs, improving patients' quality of life. In conclusion, XYS, as a traditional herbal formula, demonstrates good efficacy and safety in the treatment of depression. Compared to monotherapy with classical antidepressant therapy, the combination of XYS can yield more pronounced antidepressant effects while minimizing adverse effects. This combination therapy offers a viable long-term treatment option for improving depression. However, it is still necessary to individualize the selection and adjustment of treatment based on the specific circumstances of each patient.